# Neural Probabilistic Circuits: An Overview

**Weixin Chen**[*1]     **Simon Yu**[*2]     **Huajie Shao**[3]     **Lui Sha**[1]     **Han Zhao**[1]

[1]Department of Computer Science, University of Illinois Urbana-Champaign, Urbana, Illinois, USA
[2]Department of Electrical and Computer Engineering, University of Illinois Urbana-Champaign, Urbana, Illinois, USA
[3]Department of Computer Science, College of William and Mary, Williamsburg, Virginia, USA

## Abstract

End-to-end deep neural networks have achieved remarkable success across various domains but are often criticized for their lack of interpretability. While post hoc explanation methods attempt to address this issue, they often fail to accurately represent these black-box models, resulting in misleading or incomplete explanations. To overcome these challenges, we propose an inherently transparent model architecture called Neural Probabilistic Circuits (NPCs), which enables compositional and interpretable predictions through logical reasoning. In particular, an NPC consists of two modules: an attribute recognition model, which predicts probabilities for various attributes, and a task predictor built on a probabilistic circuit, which enables logical reasoning over recognized attributes to make class predictions. To train NPCs, we introduce a three-stage training algorithm comprising attribute recognition, circuit construction, and joint optimization. Moreover, we theoretically demonstrate that an NPC's error is upper-bounded by a linear combination of the errors from its modules. Empirical results on four benchmark datasets show that NPCs strike a balance between interpretability and performance, achieving results competitive even with those of end-to-end black-box models while providing enhanced interpretability.

## 1 INTRODUCTION

End-to-end deep neural networks (DNNs) have demonstrated remarkable success across various domains. However, many of them are black-box models containing complex operators, making it hard to interpret and understand how a decision was made. Although many efforts [Ribeiro et al., 2016, Lundberg and Lee, 2017, Selvaraju et al., 2017] have been made to explain a model's decision in a post hoc manner, Alvarez-Melis and Jaakkola [2018], Laugel et al. [2019], Rudin [2019] show that these explanations are oftentimes not reliable as the explanation model might loosely approximate the underlying model. For example, the explanation model exhibits similar performance to the black-box model but relies on entirely different features. Such discrepancy between the explanation model and the black-box model could lead to misleading explanations, *e.g.*, attributing the decision to irrelevant features or missing out important features. Misleading explanations are particularly concerning in high-stakes applications such as medical analysis and legal justice. Rather than introducing post hoc explanations to explain a black-box model, Rudin [2019] argues that one should create an interpretable model in the first place where each component is designed with a distinct purpose, facilitating an interpretable prediction.

To this end, we propose a novel model architecture called Neural Probabilistic Circuits (NPCs), which enables compositional and interpretable predictions through logical reasoning. An NPC comprises two modules: an attribute recognition model and a task predictor. Given an input image, the neural-network-based attribute recognition model produces probability vectors for various high-level, human-understandable attributes (*e.g.*, "color"), with each vector representing the likelihood of various values for the corresponding attribute. These probability vectors serve as inputs to the task predictor, which is implemented using a probabilistic circuit. Probabilistic circuits [Poon and Domingos, 2011] are a type of graphical models that learns the joint distribution over input variables, in our case, attribute variables and the class variable. During learning, probabilistic circuits embed within their structures and parameters either implicit logical rules learned from data or explicit logical rules predefined by humans. The circuits enable tractable probabilistic reasoning tasks such as joint, marginal, and conditional inferences, thereby revealing relations among the attributes and classes. By leveraging these relations,

---

[*]These authors contributed equally to this work.

*Accepted for the 8^{th} Workshop on Tractable Probabilistic Modeling at UAI* (TPM 2025).

NPCs can reason over outputs from the attribute recognition model to infer the most probable class. Specifically, the prediction score for a given class is the sum of the likelihood of each combination of attribute values weighted by their relevance to the class.

Given the compositional nature of NPCs, we propose a three-stage training algorithm. Specifically, the whole procedure involves the following stages: **1)** Attribute recognition: We begin by training the attribute recognition model within a multi-task learning framework [Caruana, 1997, Ruder, 2017]. **2)** Circuit construction: Next, we construct the circuit using two distinct approaches: i) Data-driven approach learns the circuit's structure and optimizes its parameters based on data, allowing the underlying logical rules to be embedded within the circuit. ii) Knowledge-injected approach manually designs the circuit's structure and assigns its parameters to ensure that human-predefined logical rules are explicitly encoded within the circuit. **3)** Joint optimization: Finally, the two modules are jointly optimized in an end-to-end manner to further enhance the overall model's performance on downstream tasks.

Theoretically, we demonstrate that, due to the compositional nature and the use of probabilistic circuits, NPCs exhibit a compositional error bound—the error of the overall model is upper-bounded by a linear combination of the errors from the various modules. Empirically, the results on four image classification datasets show that NPC strikes an impressive balance between interpretability and performance on downstream tasks, delivering results competitive even with those of an end-to-end deep neural network.

## 2   PRELIMINARIES

Probabilistic circuits are a class of graphical models that is used to express a joint distribution over a set of random variables $Z_{1:N}$. A probabilistic circuit $f_S$ (henceforth simply referred to as a *circuit*) consists of a rooted directed acyclic graph where leaf nodes are univariate indicators of categorical variables[1] (*i.e.*, $\mathbb{I}(Z_i = z_i)$, $z_i \in \mathcal{Z}_i$, $i \in [N]$) and internal nodes consist of sum nodes and product nodes. Each sum node computes a weighted sum of its children, and each product node computes a product of its children. In an unnormalized circuit, the root node outputs the unnormalized joint probability over variables. Any unnormalized circuit can be transformed into an equivalent, normalized circuit via weight updating Peharz et al. [2015], Zhao et al. [2015]. Hence, without loss of generality, we always assume that $f_S$ is normalized; thus, $f_S(z_{1:N}) = \Pr(Z_{1:N} = z_{1:N})$.

In circuits, the *scope* of a node is defined as the set of variables that have indicators among the node's descendants, which can be computed recursively—if $v$ is a leaf node,

say, an indicator over $Z_i$, then scope$(v) = \{Z_i\}$; otherwise, scope$(v) = \cup_{\tilde{v} \in \text{children}(v)}$ scope$(\tilde{v})$. A circuit is *smooth* iff each sum node has children with identical scope. A circuit is *decomposable* iff each product node has children with disjoint scopes. If a circuit is smooth and decomposable, then any marginal probability can be computed by setting the leaf nodes corresponding to the marginalized variables to 1. Consequently, inferences are efficient in a circuit as any joint, marginal, or conditional inference can be computed by at most two passes in a circuit. For instance, $\Pr(Z_1 = z_1 \mid Z_{2:N} = z_{2:N}) = \frac{\Pr(Z_{1:N}=z_{1:N})}{\Pr(Z_{2:N}=z_{2:N})} = \frac{f_S(Z_{1:N}=z_{1:N})}{f_S(Z_1=\emptyset, Z_{2:N}=z_{2:N})}$ where $Z_1 = \emptyset$ implies $\mathbb{I}(Z_1 = \tilde{z}_1) = 1, \forall \tilde{z}_1 \in \mathcal{Z}_1$; thus, computing a conditional probability only requires two forward processes in a circuit, each in linear time *w.r.t.* its size. We focus on smooth and decomposable circuits. A discussion on related work is deferred to Appendix A.

## 3   NEURAL PROBABILISTIC CIRCUITS

### 3.1   MODEL ARCHITECTURE AND INFERENCE

Figure 1 presents an overview of an NPC, which consists of an attribute recognition model and a task predictor. The attribute recognition model is a neural network that processes an input image to identify its high-level visual attributes, such as color and shape. The task predictor is a (normalized) probabilistic circuit that models the joint distribution over attributes and classes, embedding either implicit or explicit logical rules within its structure and parameters during learning. The circuit enables efficient probabilistic reasoning, including joint, marginal, and conditional inferences. Specifically, given a particular assignment of attributes, the circuit can infer the probability of a specific class. By leveraging these conditional dependencies alongside the probability distributions of the various attributes (*i.e.*, outputs from the attribute recognition model), NPC produces the probabilities of the image belonging to various classes.

Formally, let $X \in \mathcal{X}, A_k \in \mathcal{A}_k, Y \in \mathcal{Y}$ denote the input variable, the $k$-th attribute variable, and the class variable. The variables' instantiations are represented by $x, a_k, y$, respectively. In particular, we consider $K$ attributes, *i.e.*, $A_1, \ldots, A_K$ (or $A_{1:K}$ in short). Each attribute $A_k$ has $q_k$ possible values, *i.e.*, $|\mathcal{A}_k| = q_k$. The attribute recognition model $f(X; \theta)$ is parameterized by $\theta$. Given an input instance $x$, the model outputs $K$ probability vectors. The $k$-th probability vector, denoted as $f_k(x; \theta) \in \mathbb{R}^{q_k}$, shows the probabilities of $x$'s $k$-th attribute taking different values $a_k$, *i.e.*, $[f_k(x; \theta)]_{a_k} = \Pr_\theta (A_k = a_k \mid X = x)$. The task predictor $f_S(Y, A_{1:K}; w)$ is a probabilistic circuit with structure $S$ and parameters $w$, which models the joint distribution over $Y, A_{1:K}$. Specifically, when taking an instance of attributes $a_{1:K}$ and a class label $y$ as input, the circuit outputs the joint probability $\Pr_w(Y = y, A_{1:K} = a_{1:K})$. The circuit also supports efficient

---

[1]We mainly focus on probabilistic circuits over categorical random variables. An extension to the continuous ones is standard.

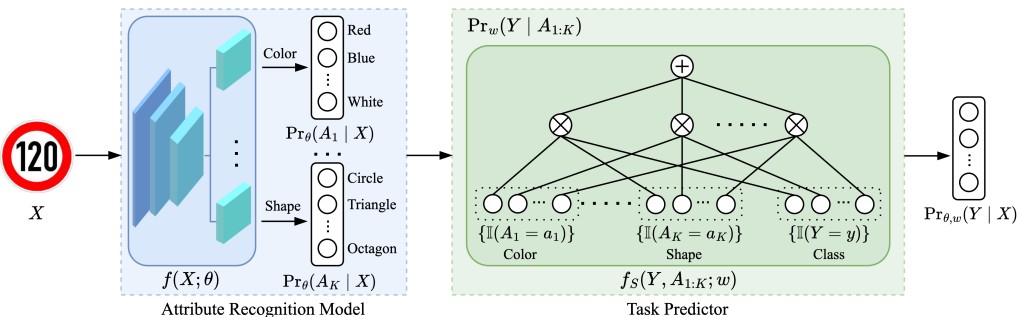

Figure 1: The compositional model architecture of an NPC. The attribute recognition model is a neural network $f(X; \theta)$ which takes an image $X$ as input and outputs $K$ probability vectors $\{\Pr_\theta(A_k \mid X)\}_{k=1}^K$. The task predictor is a probabilistic circuit $f_S(Y, A_{1:K}; w)$ taking an instance of attributes as input and providing the conditional probability $\Pr_w(Y \mid A_{1:K})$. By leveraging these relations between classes and attributes alongside the probability distributions of various attributes, NPC produces the probability vector $\Pr_{\theta,w}(Y \mid X)$.

conditional queries, e.g., $\Pr_w(Y = y \mid A_{1:K} = a_{1:K}) = f_S(y, a_{1:K}; w)/f_S(\emptyset, a_{1:K}; w)$.

Prior to describing how an NPC predicts a class, we make the following mild assumptions on the selected attributes.

**Assumption 1** (Sufficient Attributes). *Given the attributes, the class label is conditionally independent of the input, i.e.,* $Y \perp X \mid A_1, \dots, A_K$.

**Assumption 2** (Complete Information). *Given any input, all attributes are conditionally mutually independent, i.e.,* $A_1 \perp A_2 \perp \cdots \perp A_K \mid X$.

Under Assumption 1 and 2, an NPC outputs the probability of an input $x$ being a class $y$ as follows,

$$\Pr_{\theta,w}(Y = y \mid X = x)$$
$$= \sum_{a_{1:K}} \Pr_w(Y = y \mid A_{1:K} = a_{1:K}, X = x) \cdot \Pr_\theta(A_{1:K} = a_{1:K} \mid X = x)$$
$$= \sum_{a_{1:K}} \underbrace{\Pr_w(Y = y \mid A_{1:K} = a_{1:K})}_{\text{task predictor}} \cdot \underbrace{\prod_{k=1}^{K} \Pr_\theta(A_k = a_k \mid X = x)}_{\text{attribute recognition model}}.$$

The two interior terms are given by the circuit-based task predictor and the attribute recognition model, respectively.

In summary, we propose a novel model architecture for image recognition tasks. The architecture is interpretable by design, thanks to the integration of an attribute bottleneck and the probabilistic semantics of probabilistic circuits. Together, these modules enable predictions which can be interpreted using the likelihood of different attributes and the conditional dependencies between attributes and classes.

### 3.2 THREE-STAGE TRAINING ALGORITHM

**Attribute Recognition** aims to train the attribute recognition model $f(X; \theta)$ such that each attribute is recognized well. We adopt a multi-task learning framework [Zhang and Yang, 2021], where each task is to recognize a particular attribute. Specifically, we use the cross-entropy loss for each

task and assign weights to the task losses based on the size of the corresponding attribute space. That is, $\mathcal{L}_{\text{Attribute}}(\theta; D) = -\frac{1}{K} \sum_k \frac{1}{\log q_k} \left( \frac{1}{|D|} \sum_{x \in D} g_k^T(x) f_k(x; \theta) \right)$, where $f_k(x; \theta) \in \mathbb{R}^{q_k}$ and $g_k(x) \in \mathbb{R}^{q_k}$ are the output vector and the label vector corresponding to the $k$-th attribute.

**Circuit Construction** attempts to construct a probabilistic circuit $f_S(Y, A_{1:K}; w)$ that models the joint distribution over $Y, A_{1:K}$. We propose the following two distinct approaches for constructing a circuit.

In the **data-driven approach**, we *learn a circuit's structure and optimize its parameters*. Consider a training dataset $D = \{(x, y, a_{1:K})\}$. We adopt the LearnSPN Gens and Domingos [2013] to learn the structure of a circuit from the data, which recursively identifies independent groups to create product nodes, clusters data to form sum nodes, and assigns single variables as leaf nodes. Subsequently, the optimization of the circuit's weights can be framed as a maximum likelihood estimation (MLE) problem, where we employ the CCCP Zhao et al. [2016b] that iteratively applies multiplicative weight updates on $w$ to maximize the likelihood function. Overall, the learned circuit captures the underlying logical rules present in the observed data.

In the **knowledge-injected approach**, we *manually construct a circuit's structure and assign its parameters*. Domain knowledge can be represented as a set of weighted logical rules, where the weight of each rule reflects the frequency with which the rule holds true among the observed data. Consider $L$ weighted rules of the form $\{r_l := w_l \cdot (\mathbb{I}(A_1 = a_1^l) \land \dots \land \mathbb{I}(A_K = a_K^l) \land \mathbb{I}(Y = y^l))\}_{l=1}^L$. We construct a two-layer circuit to encode these rules. Building upon a set of leaf nodes that represent the indicator variables of $Y, A_{1:K}$, a layer of product nodes is built, where each product node is associated with a rule. Specifically, the $l$-th product node connects to the leaf nodes that represent the conditions in the rule $r_l$. Finally, a single sum node, which serves as the root node of the circuit, is placed above the product node layer. This sum node aggregates the outputs of all product nodes, and the weight for the $l$-th

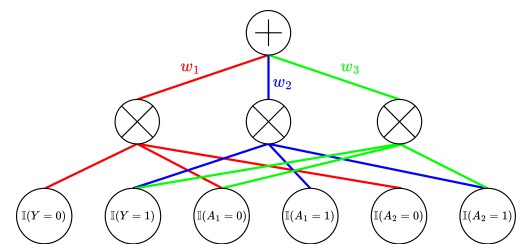

Figure 2: An illustration of a constructed two-layer circuit.

edge is assigned as $w_l$. An example of a constructed circuit is illustrated in Figure 2. Through these two steps, the human-predefined logical rules are manually encoded into the circuit's structure and parameters.

**Joint Optimization** intends to fine-tune NPCs in an end-to-end manner to further improve the performance of the overall model on downstream tasks. Thanks to the differentiability of circuits, we can optimize the overall model *w.r.t.* the loss function $\mathcal{L}_{\text{Joint}}(\theta, w; (x, y)) = -\sum_{(x,y) \in D} \log \Pr_{\theta,w}(Y = y \mid X = x)$. Specifically, we employ the stochastic gradient descent algorithm to update $\theta$, while using the projected gradient descent algorithm to update $w$ to ensure the positivity of the circuit weights.

### 3.3 THEORETICAL ANALYSIS

Given that the overall model and the attribute recognition model are discriminative while the probabilistic circuit is generative, we define the following errors to capture how closely the learned models approximate the underlying true distributions. **1) Error of the overall model**: $\epsilon_{\theta,w} := \mathbb{E}_X [d_{\text{TV}}(\Pr_{\theta,w}(Y \mid X), \Pr(Y \mid X))]$, which represents the expected total variance distance between the learned and true conditional distributions of $Y$ given $X$. **2) Error of the attribute recognition model**: $\epsilon_\theta := \mathbb{E}_X [d_{\text{TV}}(\Pr_\theta(A_{1:K} \mid X), \Pr(A_{1:K} \mid X))]$, which quantifies the expected total variation distance between the learned and true conditional distributions of the attributes $A_{1:K}$ given $X$. Additionally, we define $\epsilon_\theta^k := \mathbb{E}_X [d_{\text{TV}}(\Pr_\theta(A_k \mid X), \Pr(A_k \mid X))]$ as the error for each individual attribute $A_k$. **3) Error of the probabilistic circuit**: $\epsilon_w := d_{\text{TV}}(\Pr_w(Y, A_{1:K}), \Pr(Y, A_{1:K}))$, which measures the total variation distance between the learned and true joint distributions of $Y$ and $A_{1:K}$.

**Theorem 1** (Compositional Error). *Under Assumptions 1 and 2, the error of an NPC is bounded by a linear combination of the errors of the attribute recognition model and the circuit-based task predictor. In particular, the error of the attribute recognition model across all attributes is bounded by the sum of the errors for each attribute, i.e.,*

$$\epsilon_{\theta,w} \leqslant \epsilon_\theta + 2\epsilon_w \leqslant \sum_{k=1}^{K} \epsilon_\theta^k + 2\epsilon_w.$$

The proof is deferred to Appendix C. Theorem 1 demonstrates that the error bound of an NPC is decomposable

Table 1: Classification accuracy of NPCs and four baseline models on four benchmark datasets over five random seeds. "*" denotes uninterpretable models. The best results are highlighted in bold, while the second-best results are underlined.

| Model | MNIST-Add (%) | GTSRB (%) | CelebA (%) | AwA2 (%) |
|---|---|---|---|---|
| DNN* | $99.057 \pm 0.08$ | $\underline{99.939} \pm 0.04$ | $\mathbf{36.963} \pm 0.72$ | $\mathbf{93.351} \pm 0.17$ |
| CBM | $98.606 \pm 0.03$ | $99.810 \pm 0.04$ | $16.552 \pm 0.87$ | $82.286 \pm 0.47$ |
| CEM* | $98.740 \pm 0.10$ | $99.736 \pm 0.06$ | $25.218 \pm 0.30$ | $85.102 \pm 0.27$ |
| DCR | $94.597 \pm 2.05$ | $87.071 \pm 6.93$ | $7.055 \pm 3.04$ | $44.117 \pm 10.05$ |
| NPC(Data) | $\underline{99.171} \pm 0.11$ | $99.888 \pm 0.08$ | $\underline{33.739} \pm 0.90$ | $\underline{87.281} \pm 0.39$ |
| NPC(Know.) | $\mathbf{99.189} \pm 0.08$ | $\mathbf{99.944} \pm 0.04$ | $31.727 \pm 0.51$ | $68.519 \pm 3.54$ |

into contributions from individual modules, which accredits to its compositional nature and the incorporation of probabilistic circuits. Consequently, reducing the error of any individual module helps improve the performance of NPC.

## 4 EXPERIMENTS

**Experimental Setup.** We evaluate the model performance on four image classification datasets: MNIST-Add [Manhaeve et al., 2018], GTSRB [Stallkamp et al., 2012], CelebA [Liu et al., 2015], and AwA2 [Xian et al., 2018]. In addition to an end-to-end DNN [He et al., 2016], we select baseline models that employ a concept-based compositional architecture, such as CBM [Koh et al., 2020], CEM [Zarlenga et al., 2022], and DCR [Barbiero et al., 2023]. More details are provided in Appendix D.

**Performance Comparison.** Table 1 demonstrates that NPCs outperform all other *concept-based baseline models*. Specifically, NPC(Knowledge) achieves the best performance on MNIST-Addition and GTSRB, while NPC(Data) leads on CelebA and AwA2. Notably, NPCs achieve superior performance even compared to CEM, an uninterpretable model that relies on high-dimensional concept embeddings, highlighting NPC's effectiveness in leveraging interpretable concept probabilities for downstream classification tasks.

Remarkably, NPCs are competitive even compared with *the end-to-end DNN*, surpassing its classification accuracy on MNIST-Addition and GTSRB, while leaving small gaps on more complex datasets like CelebA and AwA2. These findings demonstrate that NPCs strike a compelling balance between interpretability and task performance, underscoring the remarkable potential of interpretable models.

## 5 CONCLUSION

In this paper, we propose Neural Probabilistic Circuits (NPCs), a novel architecture that decomposes the decision-making process into attribute recognition and logical reasoning. Our work demonstrates the potential of NPCs to enhance model interpretability and performance by integrating semantically meaningful attributes with probabilistic circuits. A discussion on limitations and future directions is deferred to Appendix B.

## Acknowledgements

We would like to extend our gratitude to Antonio Vergari for pointing out relevant literature on probabilistic circuits and knowledge compilation, and for the discussion on the relationship between NPCs and semantic probabilistic layers. We would also like to thank Rahim Khan, Tommy Tang, Alex Tanthiptham, and Trusha Vernekar for their contributions to the implementations and experiments. Finally, we thank the reviewers for their valuable suggestions, which improved the quality of this work.

WC and HZ are partially supported by an NSF IIS grant No. 2504555. HZ would like to thank the support from a Google Research Scholar Award. SY and LS are supported by the National Aeronautics and Space Administration (NASA) under Grant 80NSSC22M0070.

The views and conclusions expressed in this paper are solely those of the authors and do not necessarily reflect the official policies or positions of the supporting companies and government agencies.

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

# Neural Probabilistic Circuits: An Overview
## (Supplementary Material)

**Weixin Chen**[†1]          **Simon Yu**[*2]          **Huajie Shao**[3]          **Lui Sha**[1]          **Han Zhao**[1]

[1]Department of Computer Science, University of Illinois Urbana-Champaign, Urbana, Illinois, USA
[2]Department of Electrical and Computer Engineering, University of Illinois Urbana-Champaign, Urbana, Illinois, USA
[3]Department of Computer Science, College of William and Mary, Williamsburg, Virginia, USA

## A    RELATED WORK

In this section, we discuss several areas of research relevant to our proposed method.

**Concept Bottleneck Models and Variants**    Concept bottleneck models (CBMs) and their variants are a class of machine learning models that organize their decision-making process around high-level, human-understandable concepts, offering enhanced transparency. First introduced by Koh et al. [2020], CBMs decompose a black-box DNN into two modules: a concept recognition model, responsible for predicting various human-specified concepts, and a task predictor, which performs classifications on the predicted concepts.

Subsequent research has focused on improving these two modules. Zarlenga et al. [2022], Yeh et al. [2020], Kazhdan et al. [2020] extend the concept recognition model by representing concepts as high-dimensional embeddings rather than simple probabilities. Additionally, Mahinpei et al. [2021], Sawada and Nakamura [2022], Sarkar et al. [2022], Marconato et al. [2022] introduce unsupervised neurons into the bottleneck to enhance the model's learning capacity. While improving the performance on downstream tasks, these extensions compromise interpretability, as the dimensions within concept embeddings and the unsupervised neurons lack explicit semantic meanings. In contrast, utilizing predicted concept probabilities gives better interpretability. On the other hand, there have been recent efforts to improve the interpretability of the task predictor. Instead of using a linear layer as the task predictor, several approaches [Barbiero et al., 2023, Ciravegna et al., 2023, Rodríguez et al., 2024] design new architectures to embed logical rules and enable classifications via reasoning. For instance, Barbiero et al. [2023] propose a deep concept reasoner, while Rodríguez et al. [2024] introduce a soft decision tree as the task predictor. These approaches optimize their parameters using observed data, thus extracting the underlying logical rules inherent within the data. In comparison, architectures that directly encode human-predefined logical rules through their structure and parameters offer means to explicitly represent domain knowledge.

**Probabilistic Circuits**    Probabilistic circuits [Sánchez-Cauce et al., 2022] are rooted directed acyclic graphs designed to represent the joint distribution of a set of variables. The circuits comprise three types of nodes: 1) leaf nodes, which correspond to input variables; 2) sum nodes, which compute weighted sums of their child nodes; and 3) product nodes, which compute products of their child nodes. When satisfying the properties of decomposability and smoothness, a probabilistic circuit becomes a tractable probabilistic model, ensuring efficient inferences over various distributions [Poon and Domingos, 2011]. Specifically, joint, marginal, and conditional probabilities of input variables can be computed in at most two passes (from leaf nodes to the root node), with computational complexity linear in the size of the circuit. Consequently, probabilistic circuits combine the expressiveness of traditional graphical models with the scalability of modern deep learning frameworks.

Structure learning for probabilistic circuits aims to design structures that effectively balance expressiveness and computational efficiency. Xia et al. [2023] categorize existing structure learning methods into four types: 1) handcrafted structure learning, where structures are manually designed for specific datasets [Gens and Domingos, 2012, Poon and Domingos, 2011]; 2)

---

[*]These authors contributed equally to this work.
[†]These authors contributed equally to this work.

*Accepted for the 8th Workshop on Tractable Probabilistic Modeling at UAI  (TPM 2025).*

data-based structure learning, which uses heuristic [Adel et al., 2015, Dennis and Ventura, 2012, Gens and Domingos, 2013, Krakovna and Looks, 2016, Molina et al., 2018, Rahman and Gogate, 2016, Rooshenas and Lowd, 2014, Vergari et al., 2015] or non-heuristic algorithms [Peharz et al., 2014, Lee et al., 2014, Trapp et al., 2016, Peharz et al., 2019] to learn structures from data; 3) random structure learning, where structures are randomly generated as a flexible starting point [Peharz et al., 2019, Rashwan et al., 2016, Trapp et al., 2019]; and 4) ensemble structure learning, which combines multiple structures to improve generalization to high-dimensional data [Ventola et al., 2020]. In this paper, we utilize the first and second types of structure learning approaches to embed explicit and implicit logical rules, respectively.

Parameter learning for probabilistic circuits involves finding optimal parameters for a given structure, enabling the circuits to accurately capture the underlying probability distributions within the observed data. Parameter learning can be broadly categorized into two types: generative and discriminative. Generative parameter learning [Poon and Domingos, 2011, Peharz, 2015, Rashwan et al., 2016, Zhao et al., 2016a,b], the most common paradigm, aims to maximize the joint probabilities of all variables. The generative approach is particularly suited for tasks such as density estimation, generative modeling, and probabilistic reasoning. In contrast, discriminative parameter learning [Gens and Domingos, 2012, Adel et al., 2015, Rashwan et al., 2018] focuses on maximizing the conditional probabilities of a class variable given other variables, making it ideal for classification and regression tasks. In this paper, we adopt CCCP [Zhao et al., 2016b], a generative parameter learning approach, as it admits multiplicative parameter updates that provide a monotonic increase in the log-likelihood and lead to faster and more stable convergence.

On the other hand, some works develop probabilistic circuits parameterized by neural networks. For instance, Ahmed et al. [2022] use an amortized neural network to output the weights in circuits. To address the overfitting problem of large probabilistic circuits, Shih et al. [2021] exploit the generalization ability of a small-scale neural network and employ it to generate the parameters of a large circuit. Shao et al. [2022] aims to learn a probabilistic circuit to model the conditional distribution of target variables given input variables. In their approach, a neural network, conditioned on the input variables, is utilized to generate the circuit's parameters. This integration of neural networks enhances the expressiveness of probabilistic circuits and has been used in neuro-symbolic integration [Ahmed et al., 2022, Manhaeve et al., 2018]. In this paper, the circuit's integration with the attribute recognition model can be seen as parameterizing the input distributions of the circuit, instead of the weights of the circuit. The final predictions are a result of utilizing the outputs of a circuit through the law of total probability.

**Neuro-Symbolic Learning**    Neuro-symbolic learning integrates neural networks with symbolic representations, combining data-driven learning with symbolic reasoning to leverage the strengths of both. Variants of CBMs that embed inherent rules within the task predictor serve as an exemplifying application of neuro-symbolic learning. Beyond CBMs, this neuro-symbolic paradigm can be implemented in various other forms. One line of research focuses on designing rule-based objective functions. For instance, Badreddine et al. [2022] propose objectives that maximize the satisfiability of predefined symbolic rules over a neural network's outputs. Similarly, Xu et al. [2018], Ahmed et al. [2023] define objectives that maximize the probabilities of generating outputs aligned with symbolic rules. These objectives can also function as regularization terms alongside standard classification losses, encouraging a neural network to adhere to specific rules by optimizing its parameters accordingly. However, these works cannot ensure that the predictions are always consistent with the rules at inference time. Another line of research emphasizes the design of model architectures. For example, Manhaeve et al. [2018] propose a probabilistic-logic reasoning layer on top of the neural predicates. Different from our work, they assume a uniform label distribution and train the overall model in an end-to-end fashion, which may lead to reasoning shortcuts [Marconato et al., 2023, Bortolotti et al., 2024, 2025]. Ahmed et al. [2022] introduce a semantic probabilistic layer, a predictive layer designed for structured-output predictions, which can be seamlessly integrated into neural networks to ensure predictions align with certain symbolic constraints.

**Knowledge Compilation**    In the field of knowledge compilation [Darwiche and Marquis, 2002], previous studies have made efforts to compile PGMs [Choi et al., 2013, Chavira and Darwiche, 2005] or logical formulas [Chavira and Darwiche, 2005, Darwiche, 2002, Pipatsrisawat and Darwiche, 2008] into computationally efficient structures that support various inference tasks. In this paper, the proposed knowledge-injected parameter learning approach adopts a simple compilation strategy that compiles a set of weighted AND rules into a two-layer probabilistic circuit. Specifically, each product node corresponds to an individual AND rule, while the sum node encodes the weights associated with these rules.

# B  LIMITATIONS AND DISCUSSIONS

In this section, we discuss the limitations of NPCs from multiple perspectives, highlighting potential future directions for improvement.

**Model Architecture**  Compared to end-to-end DNNs, NPCs offer superior interpretability by decomposing a model into semantically meaningful modules, enabling humans to combine module outputs to understand the final decisions. Nevertheless, the attribute recognition model itself remains a black box, and its opaque inner workings make it difficult to ensure that its outputs truly represent the probabilities for the various attributes. For instance, the model might learn spurious correlations and incorrectly map background features, instead of actual attributes, to outputs. Future work may focus on increasing the transparency within the attribute recognition model, thereby enhancing its interpretability.

**Structure of Probabilistic Circuits**  In NPCs, the task predictor, implemented using a probabilistic circuit, is either learned using LearnSPN [Gens and Domingos, 2013] or manually constructed based on human-predefined rules. The circuit generated by LearnSPN, however, may contain an excessive number of nodes and edges, resulting in a slower inference. Alternative methods [Vergari et al., 2015, Mauro et al., 2017] may be explored to create more compact circuits for added inference efficiency. On the other hand, manually constructed circuits employ simpler structures with only two layers. While it may improve efficiency, such simplicity may limit the circuit's expressiveness, potentially degrading its performance on complex datasets like AwA2. Future work may focus on improved balancing between circuit expressiveness and structural complexity.

**Assumption of Complete Information**  The assumption of complete information in this paper assumes that attributes are conditionally independent given the input. However, recent studies suggest that this assumption limits model expressiveness [van Krieken et al., 2024] and potentially introduces spurious correlations, also known as reasoning shortcuts [Marconato et al., 2023, Bortolotti et al., 2024]. Investigating whether this issue arises in NPCs and identifying ways to address it would be a valuable direction for future research.

**Attribute Annotations**  The training of NPCs relies on the complete attribute annotations. In practice, however, such annotations are often unavailable for large-scale datasets since it would be time-consuming and labor-intensive to collect them. Yüksekgönül et al. [2023], Oikarinen et al. [2023] mitigate the need for attribute annotations by transferring concepts from other datasets or by using multimodal models to extract concepts from natural language descriptions. Therefore, it is plausible to extend NPCs to an annotation-free setting by adopting similar techniques.

**Reducing Trade-Offs between Interpretability and Task Performance**  In this paper, we show that, with the integration of attribute recognition and probabilistic circuit, NPC produces interpretable predictions for downstream tasks while achieving superior performance. Looking ahead, we believe that, by incorporating more fine-grained and diverse attributes that are semantically meaningful, along with a structure that reasons over these attributes using logical rules with increased complexity, we shall devise compositional model designs that further reduce the trade-offs between interpretability and performance of downstream tasks.

# C  PROOF FOR COMPOSITIONAL ERROR

In this section, we present a detailed proof for Theorem 1. Throughout the proof, expressions with parameters such as $\mathrm{Pr}_\theta$, $\mathrm{Pr}_w$, and $\mathrm{Pr}_{\theta,w}$ refer to probabilities learned by the models, while those without parameters represent ground-truth probabilities.

$$\epsilon_{\theta,w} = \mathbb{E}_X \left[ \frac{1}{2} \sum_y \left| \Pr_{\theta,w}(Y = y \mid X) - \Pr(Y = y \mid X) \right| \right]$$

$$\leqslant \mathbb{E}_X [ \frac{1}{2} \sum_y \sum_{a_{1:K}} | \prod_k \Pr_\theta(A_k = a_k \mid X) \cdot \Pr_w(Y = y \mid A_{1:K} = a_{1:K}) - \prod_k \Pr(A_k = a_k \mid X) \cdot \Pr_w(Y = y \mid A_{1:K} = a_{1:K})$$

$$+ \prod_k \Pr(A_k = a_k \mid X) \cdot \Pr_w(Y = y \mid A_{1:K} = a_{1:K}) - \prod_k \Pr(A_k = a_k \mid X) \cdot \Pr(Y = y \mid A_{1:K} = a_{1:K}) | ]$$

$$\leqslant \mathbb{E}_X \left[ \frac{1}{2} \sum_y \sum_{a_{1:K}} \left| \prod_k \Pr_\theta(A_k = a_k \mid X) - \prod_k \Pr(A_k = a_k \mid X) \right| \cdot \Pr_w(Y = y \mid A_{1:K} = a_{1:K}) \right] \tag{1}$$

$$+ \mathbb{E}_X \left[ \frac{1}{2} \sum_y \sum_{a_{1:K}} \prod_k \Pr(A_k = a_k \mid X) \cdot \left| \Pr_w(Y = y \mid A_{1:K} = a_{1:K}) - \Pr(Y = y \mid A_{1:K} = a_{1:K}) \right| \right]. \tag{2}$$

Thus, the upper bound of $\epsilon_{\theta,w}$ is decomposed into two terms.

We derive the upper bound for the first term, *i.e.*, Eq(1), as follows,

$$\text{Eq(1)} = \mathbb{E}_X \left[ \frac{1}{2} \sum_{a_{1:K}} \left| \prod_k \Pr_\theta(A_k = a_k \mid X) - \prod_k \Pr(A_k = a_k \mid X) \right| \right] = \mathbb{E}_X \left[ d_{\text{TV}} \left( \Pr_\theta(A_{1:K} \mid X), \Pr(A_{1:K} \mid X) \right) \right] = \epsilon_\theta$$

$$\leqslant \mathbb{E}_X \left[ \frac{1}{2} \sum_k \sum_{a_k} \left| \Pr_\theta(A_k = a_k \mid X) - \Pr(A_k = a_k \mid X) \right| \right] = \sum_k \mathbb{E}_X \left[ d_{\text{TV}} \left( \Pr_\theta(A_k = a_k \mid X), \Pr(A_k = a_k \mid X) \right) \right] = \sum_k \epsilon_\theta^k.$$

We derive the upper bound for the second term, *i.e.*, Eq(2), as follows,

$$\text{Eq(2)} = \frac{1}{2} \sum_x \sum_y \sum_{a_{1:K}} \Pr(X = x, A_{1:K} = a_{1:K}) \cdot \left| \Pr_w(Y = y \mid A_{1:K} = a_{1:K}) - \Pr(Y = y \mid A_{1:K} = a_{1:K}) \right|$$

$$= \frac{1}{2} \sum_y \sum_{a_{1:K}} \Pr(A_{1:K} = a_{1:K}) \cdot \left| \Pr_w(Y = y \mid A_{1:K} = a_{1:K}) - \Pr(Y = y \mid A_{1:K} = a_{1:K}) \right|$$

$$= \frac{1}{2} \sum_y \sum_{a_{1:K}} | \Pr(A_{1:K} = a_{1:K}) \cdot \Pr_w(Y = y \mid A_{1:K} = a_{1:K}) - \Pr_w(A_{1:K} = a_{1:K}) \cdot \Pr_w(Y = y \mid A_{1:K} = a_{1:K})$$

$$+ \Pr_w(A_{1:K} = a_{1:K}) \cdot \Pr_w(Y = y \mid A_{1:K} = a_{1:K}) - \Pr(A_{1:K} = a_{1:K}) \cdot \Pr(Y = y \mid A_{1:K} = a_{1:K}) |$$

$$\leqslant \frac{1}{2} \sum_y \sum_{a_{1:K}} \Pr_w(Y = y \mid A_{1:K} = a_{1:K}) \cdot \left| \Pr(A_{1:K} = a_{1:K}) - \Pr_w(A_{1:K} = a_{1:K}) \right|$$

$$+ \frac{1}{2} \sum_y \sum_{a_{1:K}} \left| \Pr_w(Y = y, A_{1:K} = a_{1:K}) - \Pr(Y = y, A_{1:K} = a_{1:K}) \right|$$

$$= \frac{1}{2} \sum_{a_{1:K}} \left| \Pr(A_{1:K} = a_{1:K}) - \Pr_w(A_{1:K} = a_{1:K}) \right| + d_{\text{TV}} \left( \Pr_w(Y, A_{1:K}), \Pr(Y, A_{1:K}) \right)$$

$$= \frac{1}{2} \sum_{a_{1:K}} \left| \sum_y \Pr(Y = y, A_{1:K} = a_{1:K}) - \sum_y \Pr_w(Y = y, A_{1:K} = a_{1:K}) \right| + d_{\text{TV}} \left( \Pr_w(Y, A_{1:K}), \Pr(Y, A_{1:K}) \right)$$

$$\leqslant \frac{1}{2} \sum_{a_{1:K}} \sum_y \left| \Pr(Y = y, A_{1:K} = a_{1:K}) - \Pr_w(Y = y, A_{1:K} = a_{1:K}) \right| + d_{\text{TV}} \left( \Pr_w(Y, A_{1:K}), \Pr(Y, A_{1:K}) \right)$$

$$= 2 d_{\text{TV}} \left( \Pr_w(Y, A_{1:K}), \Pr(Y, A_{1:K}) \right) = 2\epsilon_w.$$

Combining results from Eq(1) and Eq(2), we have $\epsilon_{\theta,w} \leqslant \epsilon_\theta + 2\epsilon_w \leqslant \sum_k \epsilon_\theta^k + 2\epsilon_w$.

Table 2: Model properties. "Interpretability" indicates whether the outputs produced by a concept/attribute recognition model are interpretable and whether humans can interpret the final decisions using these outputs. "Data-Driven Rules" denotes whether a model can incorporate logical rules learned from data. "Human-Predefined Rules" specifies whether a model can integrate logical rules predefined by humans. "Theoretical Guarantee" indicates whether a model provides a theoretical guarantee on the relationship between the performance of the overall model and that of its individual components.

| Model | Interpretability | Data-Driven Rules | Human-Predefined Rules | Theoretical Guarantee |
|---|---|---|---|---|
| CBM [Koh et al., 2020] | ✓ | ✗ | ✗ | ✗ |
| CEM [Zarlenga et al., 2022] | ✗ | ✗ | ✗ | ✗ |
| DCR [Barbiero et al., 2023] | ✓ | ✓ | ✗ | ✗ |
| NPC (ours) | ✓ | ✓ | ✓ | ✓ |

## D EXPERIMENTAL SETUP

In this section, we provide additional details regarding the experimental setup.

**Datasets** We evaluate the model performance on a variety of benchmark datasets. **1) MNIST-Addition:** We derive this dataset from the original MNIST dataset [LeCun et al., 1998] by following the general preprocessing steps and procedures detailed in [Manhaeve et al., 2018]. Each MNIST-Addition sample consists of two images randomly selected from the original MNIST. The digits in these images, ranging from 0 to 9, represent two attributes, with their sum serving as the class label. A total of 35,000 samples are created for MNIST-Addition. **2) GTSRB:** GTSRB [Stallkamp et al., 2012] is a dataset comprising 39,209 images of German traffic signs, with class labels indicating the type of signs. Additionally, we annotate each sample with four attributes: "color", "shape", "symbol", and "text". **3) CelebA:** CelebA [Liu et al., 2015] consists of 202,599 celebrity face images annotated with 40 binary concepts. Here, we select the 8 most balanced binary concepts[1] and group them into 5 attributes: "mouth", "face", "cosmetic", "hair", and "appearance". Following Zarlenga et al. [2022], each unique combination of concept values is treated as a group. To balance the dataset and increase its complexity, we rank these groups by the number of images they contain and pair them strategically: the group with the most images is merged with the one with the fewest, the second most with the second fewest, and so on. The above strategy results in 127 total classes. **4) AwA2:** AwA2 [Xian et al., 2018] contains 37,322 images of 50 types of animals, each annotated with 85 binary concepts. Certain concepts, such as those describing non-visual properties (*e.g.*, "fast", "domestic") or indistinctive features (*e.g.*, "chewteeth"), or those representing background information (*e.g.*, "desert" and "forest"), are excluded. After the exclusion, 29 concepts remain, which are then grouped into 4 attributes: "color", "surface", "body", and "limb". For all datasets, we split the samples into training, validation, and testing sets by a ratio of 8:1:1.

**Baselines** We select CBM [Koh et al., 2020] and several representative variants as baselines. Specifically, we choose CEM [Zarlenga et al., 2022], a method that uses high-dimensional concept embeddings as the bottleneck instead of concept probabilities, and DCR [Barbiero et al., 2023], which introduces a deep concept reasoner as the task predictor rather than relying on a simple linear layer. Additionally, we train an end-to-end DNN [He et al., 2016] as an additional baseline. It is important to note that CEM and the end-to-end DNN are not interpretable, as their components are not explicitly understandable by humans, even though they may achieve competitive performance on downstream tasks. A comparison of model properties is summarized in Table 2.

**Evaluation Metrics** We adopt the *standard classification accuracy* as the evaluation metric for the overall model.

**Model Architectures** To ensure a fair comparison with the various baselines, we consistently adopt ResNet-34 [He et al., 2016] as the backbone for all methods. Specifically, in CBM [Koh et al., 2020], the concept recognition model is based on ResNet-34, where the final layer outputs concept probabilities. The task predictor is implemented as a linear layer that takes these concept probabilities as input and outputs predicted class scores. For CEM [Zarlenga et al., 2022], the first module is implemented using ResNet-34, with its final layer being the embedding layer defined in [Zarlenga et al., 2022]. This embedding layer produces both concept embeddings and concept probabilities. The subsequent task predictor uses only the concept embeddings as input to produce the predicted class scores. In DCR [Barbiero et al., 2023], the first module is identical to that of CEM, while the task predictor is the proposed deep concept reasoner (DCR). This reasoner

---

[1]Certain concepts are excluded due to political or ethical concerns.

takes concept probabilities as input and outputs predicted class scores, during which it leverages concept embeddings to formulate logical rules. For NPC, the attribute recognition model is implemented using a multi-task learning framework. Specifically, ResNet-34, without its original final layer, serves as the feature extractor to capture common features across multiple attributes. For each attribute, a series of two linear layers acts as a dedicated task head, outputting a probability vector corresponding to the attribute.

**Human-Predefined Logical Rules** Human experience can be used to formulate specific logical rules for downstream tasks, representing valuable domain knowledge that can be integrated into models to enhance their reliability. Here, we demonstrate a two-step procedure for constructing logical rules based on a given set of observed samples $D = \{(x, y, a_{1:K})\}$. **1)** For each observed sample $(x, y, a_{1:K})$, we establish a corresponding logical rule of the form $\mathbb{I}(A_1 = a_1) \wedge \ldots \wedge \mathbb{I}(A_K = a_K) \wedge \mathbb{I}(Y = y)$. **2)** The occurrences of the logical rules derived from all observed samples are then normalized to form a weight for each rule, ensuring that the rules reflect the distribution of the observed data. Table 3 and Table 4 illustrate the rules constructed using training samples from the MNIST-Addition and GTSRB datasets. In addition to the standardized procedure introduced above, humans can also leverage their expertise to incorporate more diverse and task-specific rules beyond this form.

Table 3: Logical rules constructed using training samples from MNIST-Addition and GTSRB datasets (Part 1).

| Dataset | Logical Rules |
|---|---|
| MNIST-Addition | $\mathbb{I}(\text{Number-First} = a_1) \wedge \mathbb{I}(\text{Number-Second} = a_2) \wedge \mathbb{I}(\text{Class} = a_1 + a_2),\ a_1, a_2 \in [0, 9]$ |
| GTSRB | $\mathbb{I}(\text{Color} = \text{Red}) \wedge \mathbb{I}(\text{Shape} = \text{Circle}) \wedge \mathbb{I}(\text{Symbol} = \text{Text}) \wedge \mathbb{I}(\text{Text} = 20) \wedge \mathbb{I}(\text{Class} = \text{regulatory–maximum-speed-limit-20})$ |
| | $\mathbb{I}(\text{Color} = \text{Red}) \wedge \mathbb{I}(\text{Shape} = \text{Circle}) \wedge \mathbb{I}(\text{Symbol} = \text{Text}) \wedge \mathbb{I}(\text{Text} = 30) \wedge \mathbb{I}(\text{Class} = \text{regulatory–maximum-speed-limit-30})$ |
| | $\mathbb{I}(\text{Color} = \text{Red}) \wedge \mathbb{I}(\text{Shape} = \text{Circle}) \wedge \mathbb{I}(\text{Symbol} = \text{Text}) \wedge \mathbb{I}(\text{Text} = 50) \wedge \mathbb{I}(\text{Class} = \text{regulatory–maximum-speed-limit-50})$ |
| | $\mathbb{I}(\text{Color} = \text{Red}) \wedge \mathbb{I}(\text{Shape} = \text{Circle}) \wedge \mathbb{I}(\text{Symbol} = \text{Text}) \wedge \mathbb{I}(\text{Text} = 60) \wedge \mathbb{I}(\text{Class} = \text{regulatory–maximum-speed-limit-60})$ |
| | $\mathbb{I}(\text{Color} = \text{Red}) \wedge \mathbb{I}(\text{Shape} = \text{Circle}) \wedge \mathbb{I}(\text{Symbol} = \text{Text}) \wedge \mathbb{I}(\text{Text} = 70) \wedge \mathbb{I}(\text{Class} = \text{regulatory–maximum-speed-limit-70})$ |
| | $\mathbb{I}(\text{Color} = \text{Red}) \wedge \mathbb{I}(\text{Shape} = \text{Circle}) \wedge \mathbb{I}(\text{Symbol} = \text{Text}) \wedge \mathbb{I}(\text{Text} = 80) \wedge \mathbb{I}(\text{Class} = \text{regulatory–maximum-speed-limit-80})$ |
| | $\mathbb{I}(\text{Color} = \text{Red}) \wedge \mathbb{I}(\text{Shape} = \text{Circle}) \wedge \mathbb{I}(\text{Symbol} = \text{Text}) \wedge \mathbb{I}(\text{Text} = 100) \wedge \mathbb{I}(\text{Class} = \text{regulatory–maximum-speed-limit-100})$ |
| | $\mathbb{I}(\text{Color} = \text{Red}) \wedge \mathbb{I}(\text{Shape} = \text{Circle}) \wedge \mathbb{I}(\text{Symbol} = \text{Text}) \wedge \mathbb{I}(\text{Text} = 120) \wedge \mathbb{I}(\text{Class} = \text{regulatory–maximum-speed-limit-120})$ |
| | $\mathbb{I}(\text{Color} = \text{White}) \wedge \mathbb{I}(\text{Shape} = \text{Circle}) \wedge \mathbb{I}(\text{Symbol} = \text{Text}) \wedge \mathbb{I}(\text{Text} = 80) \wedge \mathbb{I}(\text{Class} = \text{regulatory–end-of-maximum-speed-limit-80})$ |
| | $\mathbb{I}(\text{Color} = \text{Red}) \wedge \mathbb{I}(\text{Shape} = \text{Circle}) \wedge \mathbb{I}(\text{Symbol} = \text{Cat-Two}) \wedge \mathbb{I}(\text{Text} = \text{Undefined}) \wedge \mathbb{I}(\text{Class} = \text{regulatory–no-overtaking})$ |
| | $\mathbb{I}(\text{Color} = \text{Red}) \wedge \mathbb{I}(\text{Shape} = \text{Circle}) \wedge \mathbb{I}(\text{Symbol} = \text{Car-Truck}) \wedge \mathbb{I}(\text{Text} = \text{Undefined}) \wedge \mathbb{I}(\text{Class} = \text{regulatory–no-overtaking-by-heavy-goods-vehicles})$ |
| | $\mathbb{I}(\text{Color} = \text{Red}) \wedge \mathbb{I}(\text{Shape} = \text{Triangle}) \wedge \mathbb{I}(\text{Symbol} = \text{Arrow-Up}) \wedge \mathbb{I}(\text{Text} = \text{Undefined}) \wedge \mathbb{I}(\text{Class} = \text{warning–crossroads})$ |
| | $\mathbb{I}(\text{Color} = \text{White}) \wedge \mathbb{I}(\text{Shape} = \text{Diamond}) \wedge \mathbb{I}(\text{Symbol} = \text{Undefined}) \wedge \mathbb{I}(\text{Text} = \text{Undefined}) \wedge \mathbb{I}(\text{Class} = \text{regulatory–priority-road})$ |
| | $\mathbb{I}(\text{Color} = \text{Red}) \wedge \mathbb{I}(\text{Shape} = \text{Triangle}) \wedge \mathbb{I}(\text{Symbol} = \text{Undefined}) \wedge \mathbb{I}(\text{Text} = \text{Undefined}) \wedge \mathbb{I}(\text{Class} = \text{regulatory–yield})$ |
| | $\mathbb{I}(\text{Color} = \text{Red}) \wedge \mathbb{I}(\text{Shape} = \text{Octagon}) \wedge \mathbb{I}(\text{Symbol} = \text{Text}) \wedge \mathbb{I}(\text{Text} = \text{Stop}) \wedge \mathbb{I}(\text{Class} = \text{regulatory–stop})$ |
| | $\mathbb{I}(\text{Color} = \text{Red}) \wedge \mathbb{I}(\text{Shape} = \text{Circle}) \wedge \mathbb{I}(\text{Symbol} = \text{Undefined}) \wedge \mathbb{I}(\text{Text} = \text{Undefined}) \wedge \mathbb{I}(\text{Class} = \text{regulatory–road-closed-to-vehicles})$ |
| | $\mathbb{I}(\text{Color} = \text{Red}) \wedge \mathbb{I}(\text{Shape} = \text{Circle}) \wedge \mathbb{I}(\text{Symbol} = \text{Truck}) \wedge \mathbb{I}(\text{Text} = \text{Undefined}) \wedge \mathbb{I}(\text{Class} = \text{regulatory–no-heavy-goods-vehicles})$ |
| | $\mathbb{I}(\text{Color} = \text{Red}) \wedge \mathbb{I}(\text{Shape} = \text{Circle}) \wedge \mathbb{I}(\text{Symbol} = \text{Slash}) \wedge \mathbb{I}(\text{Text} = \text{Undefined}) \wedge \mathbb{I}(\text{Class} = \text{regulatory–no-entry})$ |
| | $\mathbb{I}(\text{Color} = \text{Red}) \wedge \mathbb{I}(\text{Shape} = \text{Triangle}) \wedge \mathbb{I}(\text{Symbol} = \text{Exclamation-Mark}) \wedge \mathbb{I}(\text{Text} = \text{Undefined}) \wedge \mathbb{I}(\text{Class} = \text{warning–other-danger})$ |
| | $\mathbb{I}(\text{Color} = \text{Red}) \wedge \mathbb{I}(\text{Shape} = \text{Triangle}) \wedge \mathbb{I}(\text{Symbol} = \text{Arrow-Left}) \wedge \mathbb{I}(\text{Text} = \text{Undefined}) \wedge \mathbb{I}(\text{Class} = \text{warning–curve-left})$ |
| | $\mathbb{I}(\text{Color} = \text{Red}) \wedge \mathbb{I}(\text{Shape} = \text{Triangle}) \wedge \mathbb{I}(\text{Symbol} = \text{Arrow-Right}) \wedge \mathbb{I}(\text{Text} = \text{Undefined}) \wedge \mathbb{I}(\text{Class} = \text{warning–curve-right})$ |
| | $\mathbb{I}(\text{Color} = \text{Red}) \wedge \mathbb{I}(\text{Shape} = \text{Triangle}) \wedge \mathbb{I}(\text{Symbol} = \text{Arrow-Consecutive-Turns}) \wedge \mathbb{I}(\text{Text} = \text{Undefined}) \wedge \mathbb{I}(\text{Class} = \text{warning–double-curve-first-left})$ |
| | $\mathbb{I}(\text{Color} = \text{Red}) \wedge \mathbb{I}(\text{Shape} = \text{Triangle}) \wedge \mathbb{I}(\text{Symbol} = \text{Bump}) \wedge \mathbb{I}(\text{Text} = \text{Undefined}) \wedge \mathbb{I}(\text{Class} = \text{warning–uneven-road})$ |
| | $\mathbb{I}(\text{Color} = \text{Red}) \wedge \mathbb{I}(\text{Shape} = \text{Triangle}) \wedge \mathbb{I}(\text{Symbol} = \text{Car}) \wedge \mathbb{I}(\text{Text} = \text{Undefined}) \wedge \mathbb{I}(\text{Class} = \text{warning–slippery-road-surface})$ |

Table 4: Logical rules constructed using training samples from MNIST-Addition and GTSRB datasets (Part 2).

| Dataset | Logical Rules |
|---|---|
| GTSRB | $\mathbb{I}$(Color = Red) $\wedge$ $\mathbb{I}$(Shape = Triangle) $\wedge$ $\mathbb{I}$(Symbol = Road-Narrows) $\wedge$ $\mathbb{I}$(Text = Undefined) $\wedge$ $\mathbb{I}$(Class = warning–road-narrows-right) |
| | $\mathbb{I}$(Color = Red) $\wedge$ $\mathbb{I}$(Shape = Triangle) $\wedge$ $\mathbb{I}$(Symbol = Roadworks) $\wedge$ $\mathbb{I}$(Text = Undefined) $\wedge$ $\mathbb{I}$(Class = warning–roadworks) |
| | $\mathbb{I}$(Color = Red) $\wedge$ $\mathbb{I}$(Shape = Triangle) $\wedge$ $\mathbb{I}$(Symbol = Traffic-Signal) $\wedge$ $\mathbb{I}$(Text = Undefined) $\wedge$ $\mathbb{I}$(Class = warning–traffic-signals) |
| | $\mathbb{I}$(Color = Red) $\wedge$ $\mathbb{I}$(Shape = Triangle) $\wedge$ $\mathbb{I}$(Symbol = Person) $\wedge$ $\mathbb{I}$(Text = Undefined) $\wedge$ $\mathbb{I}$(Class = warning–pedestrians-crossing) |
| | $\mathbb{I}$(Color = Red) $\wedge$ $\mathbb{I}$(Shape = Triangle) $\wedge$ $\mathbb{I}$(Symbol = Person-Two) $\wedge$ $\mathbb{I}$(Text = Undefined) $\wedge$ $\mathbb{I}$(Class = warning–children) |
| | $\mathbb{I}$(Color = Red) $\wedge$ $\mathbb{I}$(Shape = Triangle) $\wedge$ $\mathbb{I}$(Symbol = Bicycle) $\wedge$ $\mathbb{I}$(Text = Undefined) $\wedge$ $\mathbb{I}$(Class = warning–bicycles-crossing) |
| | $\mathbb{I}$(Color = Red) $\wedge$ $\mathbb{I}$(Shape = Triangle) $\wedge$ $\mathbb{I}$(Symbol = Ice-or-Snow) $\wedge$ $\mathbb{I}$(Text = Undefined) $\wedge$ $\mathbb{I}$(Class = warning–ice-or-snow) |
| | $\mathbb{I}$(Color = Red) $\wedge$ $\mathbb{I}$(Shape = Triangle) $\wedge$ $\mathbb{I}$(Symbol = Deer) $\wedge$ $\mathbb{I}$(Text = Undefined) $\wedge$ $\mathbb{I}$(Class = warning–wild-animals) |
| | $\mathbb{I}$(Color = White) $\wedge$ $\mathbb{I}$(Shape = Circle) $\wedge$ $\mathbb{I}$(Symbol = Undefined) $\wedge$ $\mathbb{I}$(Text = Undefined) $\wedge$ $\mathbb{I}$(Class = regulatory–end-of-prohibition) |
| | $\mathbb{I}$(Color = Blue) $\wedge$ $\mathbb{I}$(Shape = Circle) $\wedge$ $\mathbb{I}$(Symbol = Arrow-Right) $\wedge$ $\mathbb{I}$(Text = Undefined) $\wedge$ $\mathbb{I}$(Class = regulatory–turn-right-ahead) |
| | $\mathbb{I}$(Color = Blue) $\wedge$ $\mathbb{I}$(Shape = Circle) $\wedge$ $\mathbb{I}$(Symbol = Arrow-Left) $\wedge$ $\mathbb{I}$(Text = Undefined) $\wedge$ $\mathbb{I}$(Class = regulatory–turn-left-ahead) |
| | $\mathbb{I}$(Color = Blue) $\wedge$ $\mathbb{I}$(Shape = Circle) $\wedge$ $\mathbb{I}$(Symbol = Arrow-Up) $\wedge$ $\mathbb{I}$(Text = Undefined) $\wedge$ $\mathbb{I}$(Class = regulatory–go-straight) |
| | $\mathbb{I}$(Color = Blue) $\wedge$ $\mathbb{I}$(Shape = Circle) $\wedge$ $\mathbb{I}$(Symbol = Arrow-Up-and-Right) $\wedge$ $\mathbb{I}$(Text = Undefined) $\wedge$ $\mathbb{I}$(Class = regulatory–go-straight-or-turn-right) |
| | $\mathbb{I}$(Color = Blue) $\wedge$ $\mathbb{I}$(Shape = Circle) $\wedge$ $\mathbb{I}$(Symbol = Arrow-Up-and-Left) $\wedge$ $\mathbb{I}$(Text = Undefined) $\wedge$ $\mathbb{I}$(Class = regulatory–go-straight-or-turn-left) |
| | $\mathbb{I}$(Color = Blue) $\wedge$ $\mathbb{I}$(Shape = Circle) $\wedge$ $\mathbb{I}$(Symbol = Arrow-Down-Right) $\wedge$ $\mathbb{I}$(Text = Undefined) $\wedge$ $\mathbb{I}$(Class = regulatory–keep-right) |
| | $\mathbb{I}$(Color = Blue) $\wedge$ $\mathbb{I}$(Shape = Circle) $\wedge$ $\mathbb{I}$(Symbol = Arrow-Down-Left) $\wedge$ $\mathbb{I}$(Text = Undefined) $\wedge$ $\mathbb{I}$(Class = regulatory–keep-left) |
| | $\mathbb{I}$(Color = Blue) $\wedge$ $\mathbb{I}$(Shape = Circle) $\wedge$ $\mathbb{I}$(Symbol = Arrow-Roundabout) $\wedge$ $\mathbb{I}$(Text = Undefined) $\wedge$ $\mathbb{I}$(Class = regulatory–roundabout) |
| | $\mathbb{I}$(Color = White) $\wedge$ $\mathbb{I}$(Shape = Circle) $\wedge$ $\mathbb{I}$(Symbol = Car-Two) $\wedge$ $\mathbb{I}$(Text = Undefined) $\wedge$ $\mathbb{I}$(Class = regulatory–end-of-no-overtaking) |
| | $\mathbb{I}$(Color = White) $\wedge$ $\mathbb{I}$(Shape = Circle) $\wedge$ $\mathbb{I}$(Symbol = Car-Truck) $\wedge$ $\mathbb{I}$(Text = Undefined) $\wedge$ $\mathbb{I}$(Class = regulatory–end-of-no-overtaking-by-heavy-goods-vehicles) |