# OpenReview forum: "Neural Probabilistic Circuits: An Overview"
_auai.org/UAI/2025/Workshop/TPM — TPM 2025_

### Official Review · Reviewer_eAau · 2025-06-13
**Review of Submission9: A well-motivated application of PCs for concept-based predictions.**

**Rating:** 3

**Review:**

**Short summary:** The paper introduces neural probabilistic circuits (NPCs), a concept-based architecture aiming to provide interpretable model predictions. NPCs consist of two modules, namely a neural concept extractor and a task predictor encoded by a probabilistic circuit (PC), specifying a joint distribution over concepts and classification labels that can be tractably marginalized and conditioned. Besides learning the PC and its structure from data, the paper shows how to construct it from human-specified rules to guide the prediction. Most interestingly, the paper gives an upper bound on the error of the full architecture, showing that it depends linearly on the errors of the individual components. In the empirical evaluation on four image classification tasks, NPCs seem to be competitive in terms of accuracy to other concept-based architectures as well as a black-box DNN.

**Strengths:** The paper is for the most part very well-written and easy to understand. Important concepts such as probabilistic circuits and their relevant properties are explained clearly and concisely. Figure 1 gives a good overview of the general setup and the proposed model architecture, while Figure 2 helped me understand the structure of the constructed circuits for human-defined rules. The motivation for NPCs is clearly presented and the initial empirical results are promising. Moreover, I find it noteworthy that the authors also study the error of NPCs theoretically and provide a detailed proof of their derived bound in the appendix.

**Weaknesses / Suggestions for improvement:**
From my perspective, the introduction of the paper could be shortened (to e.g. only the first page) in favour of including relevant details regarding the learning, experiments, and baselines. For instance, the concrete formula of the multi-task loss used to train the attribute recognition model does not seem to appear in the main paper nor the appendix. Moreover, the number and source of the human-specified rules used in the experiments seem to be missing. I was looking for this information due to the comparatively low performance of the knowledge-based approach on AwA2. Moreover, there is little information on how the baselines the model is compared against were trained. Lastly, a short discussion of the limitations of the method would have been a nice addition to the paper in my opinion. For instance, the method seems to rely on full annotations of the attributes for training. This is common for concept bottleneck models but seems worth a mention when comparing to methods that do not require the annotation, such as the black-box DNN. Also, the plausibility and practical implications of the assumptions made in Section 3.1 are currently not discussed.
On a first pass over the paper, I was also briefly confused whether $A_k$ and $a_k$ were referring to the full vector of values for the k-th attribute (e.g. for shape ('circle', 'traingle', 'octagon', ...)) or just the single value of the attribute associated with the current image (e.g., just 'circle'). To avoid this potential confusion for other readers, I would suggest explicitly clarifying this distinction when introducing the notation or highlighting the "active" concepts in the example in Figure 1.

Despite the suggestions for improvement mentioned above, I do think this paper is worth accepting since it shows a well-motivated use of PCs for building interpretable models, with promising initial empirical results and an interesting theoretical error analysis.

---

### Official Review · Reviewer_KGSm · 2025-06-16
**good paper**

**Rating:** 3

**Review:**

# Summary

The paper introduces Neural Probabilistic Circuits, which combine a pre-trained attribute prediction neural network with a probabilistic circuit (either learned from data or compiled from prior knowledge).  This setup is capable of solving prediction tasks in an interpretable manner (but see my questions). NPCs are compared to several concept-based classifiers and found to outperform them in several tasks.

# Strengths

- Easy to read, all prerequisites are provided in sufficient detail.

- The proposed method is conceptually simple but it combines known elements into a well designed package.

- The experiments are convincing and the results quite positive.

# Weaknesses

- From an architectural perspective, NPC(Knowledge) are *very* close to a propositional version of DeepProbLog.

To the best of my understanding, the only difference is that the neural predicates are pre-trained in the Attribute Recognition stage. Mind you, this is a critical difference, as it can help substantially lessen the negative effects of Reasoning Shortcuts (see Marconato et al., "Not all neuro-symbolic concepts are born equal", 2023 for NPC(Knowledge) and Bortolotti et al, "Shortcuts and Identifiability in Concept-based Models from a Neuro-Symbolic Lens", 2025 for NPC(Data)). This is also why I believe NPCs are indeed more interpretable than DeepProbLog.

Still, and this is my main point, it would make sense to mention these similarities and differences in the paper.

- Interpretability is not evaluated explicitly, e.g., with a user study.  Just providing supervision for the concepts is not enough to avoid, e.g., concept leakage (see Mahinpei's work, which you cite).  This begs the question of how interpretable are the attributes that NPCs learn in practice.

- More generally, the related work (and ideally the experiments) should include a brief discussion of neuro-symbolic approaches to constrained classification, which fall in the same ballparck as NPCs.  I am thinking of works by Eleonora Giunchiglia (hierarchical multiclass CNNs) and Vaishak Belle (multiplexnet).  These are not probabilistic in nature, but they are very relevant.

- I am a bit puzzled by the experimental results.  Specifically, the fact that NPC(Data) outperforms NPC(Knowledge) in two tasks seems to indicate that the knowledge in those tasks is not ideal.  I realize this is just an overview, but -- as a reader and fellow researcher -- I'd like to know what is happening there exactly.

- I did not check whether the results for CBMs, CEMs and DCR match those in the literature - but I find their performance a bit lackluster. Am I mistaken?

- Finally, and this is an important but open ended question: how does one identify what concepts are responsible for a prediction? Having a bottleneck helps with interpretability, but it does not automatically guarantee being able to obtain faithful local explanations. The main advantage of CBMs is that the inference layer is linear, so it is easy to work out what attributes matter the most (if you squint your eyes enough). PCs, on the other hand, can be quite contrived, especially when learned from data or compiled from complex knowledge.